# OpenReview forum: "Curiosity-driven Red-teaming for Large Language Models"
_ICLR.cc/2024/Conference — ICLR 2024 poster_

### Official Review · Reviewer_NqgB · 2023-10-29

**Soundness:** 3 good
**Presentation:** 3 good
**Contribution:** 3 good
**Rating:** 8
**Confidence:** 3

**Summary:**

This paper points out that the current red teaming approaches based on RL can not generate test cases with high diversity. To address this, the authors propose a curiosity-driven exploration method to train the read team models. This approach jointly maximizes the red team effectiveness and also a diversity reward, where the authors tried different metrics.  Experimental results show that the proposed approach can not only maintain the effectiveness but also increase the test case diversity, compared with previous RL-based approaches. However, the experiments are mainly based on small models GPT2 with 137M parameters, making the results and the claim of red-teaming for "Large Language Models"less convincing.

**Strengths:**

This paper introduces a novel approach to training models that generate more diverse red team test cases. A diversity reward is introduced to achieve that, by encouraging the red team models to explore more diverse cases. The authors test different ways to define the rewards.
Results on GPT2 models show that the approach can indeed increase the test case diversity while maintaining effectiveness.

**Weaknesses:**

The experiments chose GPT2 as the target model in the main results, making the results not too convincing. I recommend testing on a wider range of LLMs, including proprietary models like ChatGPT and open-sourced models like LLaMA-2-chat (the user has already done but there are not enough results and details) and Vicuna.

**Questions:**

1. Are the RoBERTa hate speeh classifier strong enough to detect the toxic generations?
2. Why not try "real" LLMs like proprietary models like ChatGPT and open-sourced models like LLaMA-2-chat and Vicuna? I do not think the results are GPT2 with 137M parameters are convincing enough.

---

> ### Author Response · Authors · 2023-11-17
> **response**
>
> We thank the reviewer for appreciating the novelty of our work. We’ve added
>
> 1. Experiments on Vicuna and ChatGPT (Appendix C.3)
> 2. Analysis on the toxicity classifier  (Appendix C.1)
>
> to the manuscript. The following is our detailed response.
>
> > The experiments chose GPT2 as the target model in the main results, making the results not too convincing. I recommend testing on a wider range of LLMs, including proprietary models like ChatGPT and open-sourced models like LLaMA-2-chat (the user has already done but there are not enough results and details) and Vicuna.
> >
>
> > Why not try "real" LLMs like proprietary models like ChatGPT and open-sourced models like LLaMA-2-chat and Vicuna? I do not think the results are GPT2 with 137M parameters are convincing enough.
> >
>
> **Answer:**
>
> In addition to GPT2, our main results also include experiments against **Dolly-7B and LLaMA2-7B-chat-hf**, demonstrating our method is able to create diverse and effective testcases.
>
> We’ve added the results of red teaming against Vicuna-7B and ChatGPT (gpt3.5-turbo) in Section C.3 in the Appendix, showing that **our method is also effective at achieving diverse and effective testcases** against these models. Our results show the following:
>
> 1. **Higher Diversity:** Our approach yields more diverse test cases and target model’s responses, as shown in (Figures 9(i)(c,d,e,f) and 9(ii)(c,d,e,f)).
> 2. **More Unique Test Cases:** Figures 9(i)(b) and 9(ii)(b) illustrate that our RL+Curiosity approach generated around 50,000 unique test cases resulting in toxic responses surpassing a 0.9 toxicity threshold. This number is significantly higher than those produced by the baselines, RL+TDiv and RL. It's important to note, as Figures 9(i)(a) and 9(ii)(a) indicate, that while RL+TDiv led to a higher proportion of toxic responses, the overall lower number of unique test cases indicates an issue of lack of diversity, with many cases being exactly identical based on string comparison.
>
> > 1. Are the RoBERTa hate speeh classifier strong enough to detect the toxic generations?
> >
>
> **Answer:**
>
> Yes. To show that, we compared the RoBERTa model’s classification with GPT-4’s classification. We sampled 3000 target model’s responses each for all red-teaming methods and prompted GPT-4 to predict whether the response was toxic or not. Table 7(iii) in the Appendix showed the precision, recall, and F1-score with GPT-4’s predictions as ground truth and RoBERTa as an estimation. The table shows that that F1 scores are all close to one, which indicates that RoBERTa’s predictions are similar to GPT-4’s predictions. We also added the confusion matrix in Figure 7(ii) in the Appendix for reference.

---

> > ### Author Response · Authors · 2023-11-21
> >
> > Dear Reviewer NqgB,
> >
> > Since the discussion period is going to end soon, we would like to know if the reviewer has follow-up questions. We are happy to answer any follow-up questions. Thanks!

---

### Official Review · Reviewer_TuLd · 2023-11-01

**Soundness:** 4 excellent
**Presentation:** 4 excellent
**Contribution:** 3 good
**Rating:** 8
**Confidence:** 3

**Summary:**

- The paper proposes an rl-based red teaming method that can explore novel test cases by adapting a curiosity-driven exploration technique.
- The proposed method avoids redundant test cases by utilizing novelty reward during the rl optimization procedure.
- The empirical results show that the proposed curiosity-driven red teaming method achieves superior performance in both the red team accuracy and the diversity compared to the baseline red teaming methods.

**Strengths:**

- The idea is intuitive and easy to understand.
- The writing is clear.
- The authors provide experimental results with two diversity measures, 1-self-bleu and 1-cosine similarity.
- The empirical results are amazing. The proposed method shows higher red teaming performance than the rl-based red teaming methods while maintaining a diversity score comparable to the zero-shot baseline method.

**Weaknesses:**

- Since the paper handles the problem of red-teaming which is possible to make ethical issues in society, I think it would be better to contain subsections for ethical comments in the red-teaming.
- The empirical results are limited to the text-to-text generation models.
- The proposed automated red-teaming method is based on the pre-trained offensiveness classifier. As [1] raised, there exists a risk of discovering test cases that over-fit the red team classifier, resulting in false positive test cases. But there isn't any analysis about this risk.

[1] Query-Efficient Black-Box Red Teaming via Bayesian Optimization, Lee et al., ACL 2023.

**Questions:**

- Can you provide simple experimental results on text-to-image models like stable-diffusion?
- Can you provide an analysis of the classifier overfitting problem? For example, you may provide a confusion matrix for each method.
- Can you provide experimental results on large language models such as the text-davinci-003 model or gpt-3.5-turbo prompted chatbots?
- It is just a curiosity question. In a realistic scenario, each query on the target model usually incurs costs. Hence, the number of model queries during the red teaming process can be an important factor for the overall cost of the method. Can you provide the number of queries used during the end-to-end process of RL+"curiosity"? Also, can you estimate the price when we red-team gpt4 api using RL +"curiosity"?

**Details Of Ethics Concerns:**

The paper has the potential to be maliciously utilized by attackers to make adversarial prompts for publicly released AI models. Hence, some ethical comments should be added to the paper.

---

> ### Author Response · Authors · 2023-11-17
> **response**
>
> We thank the reviewer for appreciating our writing and empirical results. We added
>
> 1. Ethic statement (after the Discussion section)
> 2. Experiments on ChatGPT and Vicuna (Appendix C.3)
> 3. Analysis on over-fitting to classifier (Appendix C.1)
>
> to the manuscript. Also, we are running experiments on text-to-image experiment. Our responses are detailed in the following.
>
> > Ethical comments
> >
>
> **Answer:**
>
> We’ve added an ethic section after the discussion section and would like to know if the reviewer thinks this addresses potential ethic concerns.
>
> > The empirical results are limited to the text-to-text generation models.
> >
>
> > Can you provide simple experimental results on text-to-image models like stable-diffusion?
> >
>
> **Answer:**
>
> Our framework is applicable to text-to-image red-teaming. We are running the experiments and will do our best to add these results during rebuttal.
>
> > The proposed automated red-teaming method is based on the pre-trained offensiveness classifier. As [1] raised, there exists a risk of discovering test cases that over-fit the red team classifier, resulting in false positive test cases. But there isn't any analysis about this risk.
> >
>
> **Answer:**
>
> We added a new analysis in Appendix C.3, showing that the toxic response elicited by our red-teaming method **generalize to other toxicity classifiers** also.
>
> **High Correlation of Toxicity Predictions Across Classifiers:** Table 7(i) shows **Pearson Correlation Coefficients (PCC)** of the toxicity probabilities predictions between RoBERTa and two other classifiers, on the responses of the target model Dolly-7B. The table shows that PCC is near one, indicating that a strong positive correlation between RoBERTa and other classifiers in toxicity prediction. This implies that when RoBERTa predicts higher toxicity probability, the other models are also likely to show similar increases in their toxicity probability predictions.
>
> **Similar to GPT-4’s predictions:** Table 7(iii) in Appendix C.3 compared RoBERTa's classification accuracy to GPT-4 by analyzing 3000 responses from each red-teaming method. GPT-4 evaluated these responses for toxicity. Using GPT-4's predictions as the ground truth, we assessed RoBERTa's precision, recall, and F1-score. The results, with F1 scores nearing one, suggest a high similarity between RoBERTa's and GPT-4's predictions. We also added the confusion matrix in Figure 7(ii) the Appendix C.3 for reference.
>
> Note that we compared RoBERTa with GPT-4 using a confusion matrix and F1 score rather than Pearson Correlation Coefficient (PCC), as GPT-4 cannot predict probabilities. Attempts to prompt GPT-4 for probabilistic outputs, using various phrasings, were unsuccessful; the model typically responded that it could not provide numerical probabilities.
>
> > Can you provide experimental results on large language models such as the text-davinci-003 model or gpt-3.5-turbo prompted chatbots?
> >
>
> **Answer:**
>
> We’ve added the results of red teaming against Vicuna-7B and ChatGPT (gpt3.5-turbo) in Section C.3 in the Appendix, showing that **our method is also effective at generating diverse and effective testcases** against these models. Our results showed the following facts:
>
> 1. **Higher Diversity:** Our approach yields more diverse test cases and target model’s responses, as shown in (Figures 9(i)(c,d,e,f) and 9(ii)(c,d,e,f)).
> 2. **More Unique Test Cases:** Figures 9(i)(b) and 9(ii)(b) illustrate that our RL+Curiosity approach generated around 50,000 unique test cases resulting in toxic responses surpassing a 0.9 toxicity threshold. This number is significantly higher than those produced by the baselines, RL+TDiv and RL. It's important to note, as Figures 9(i)(a) and 9(ii)(a) indicate, that while RL+TDiv led to a higher proportion of toxic responses, the overall lower number of unique test cases indicates an issue of lack of diversity, with many cases being exactly identical based on string comparison.
>
> > It is just a curiosity question. In a realistic scenario, each query on the target model usually incurs costs. Hence, the number of model queries during the red teaming process can be an important factor for the overall cost of the method. Can you provide the number of queries used during the end-to-end process of RL+"curiosity"? Also, can you estimate the price when we red-team gpt4 api using RL +"curiosity"?
> >
>
> **Answer:**
>
> **Number of queries:** We used 40K queries for instruction following experiments (Section 4.3) and 100K queries for text continuation experiments (Section 4.2). We determine both numbers based on when the combined rewards (i.e., toxicity probability + curiosity + entropy bonus) of RL+Curiosity (our method) stop growing.
>
> **Price of red-teaming GPT-4:** $143.999. We did the calculation by assuming the maximum amount of input and output tokens are all 40 and 40K queries and based on the GPT-4’s pricing info in https://openai.com/pricing.

---

> > ### Comment · Reviewer_TuLd · 2023-11-20
> >
> > Thank you for your kind responses. Most of my inquiries were addressed in the authors' responses. I raised my overall score.

---

> > > ### Author Response · Authors · 2023-11-23
> > >
> > > We thank the reviewer for raising the score. In the meantime, we also added the text-to-image red-teaming experiment with the stable-diffusion model in Appendix C4. The results showed that our approach achieves higher diversity than the baselines and matches quality with them.

---

### Official Review · Reviewer_ZhdE · 2023-11-05

**Soundness:** 3 good
**Presentation:** 4 excellent
**Contribution:** 3 good
**Rating:** 8
**Confidence:** 3

**Summary:**

- The paper proposes to add an entropy bonus and novelty reward in addition to the standard RL objective to maximize toxicity in the target LLM’s responses. Two variations of the novelty reward are proposed: A SelfBLEU novelty reward and a cosine similarity novelty reward.
- The proposed method of red-teaming is evaluated in a text continuation task(evaluated on GPT2), an instruction following task (GPT2-alpaca and Dolly-v2-7B). The method is also evaluated on LLMs fine-tuned with human preference (LLaMA2-7b-chat-hf)
- The authors conduct ablations on the various KL penalty coefficients, sampling temperature, and the necessity of each reward. Studies on the hyperparameter choices are shown in the appendix.

**Strengths:**

- The paper tackles an important problem and contributions are clear and well presented.

- The method (RL+Curiosity) is simple and sound, borrowing from known methods in the RL literature.

- Experiments cover a good number of models, in a number of settings, and performance of RL+Curiosity is noticeably better than baselines.

- Ablation studies alleviate concerns that test case diversity is not simply fixed by increasing sampling temperature or modifying the KL penalty.

**Weaknesses:**

- The authors choose an objective of maximizing test case novelty but do not show how their method influences target LLM response novelty.

- In A.7, you mention you sample 100 sets of 100 test cases and calculate both diversity metrics across each subset. What is the test case size for a select number of thresholds in your experiments, and why is this sampling method used?

- This is not too important but it is worth mentioning that the novelty of the method is somewhat limited. The cosine similarity reward is very similar to the RL+TDiv baseline, except it is applied to the test cases.

In general, the paper tackles a relevant problem and is well motivated and presented. I have some limited concerns (see above) and would like to hear the author's rebuttal, after which I may modify my score.

**Questions:**

- Could you clarify what number of test cases exceed each toxicity threshold for RL+Curiosity for the experiments in section 4.2 and 4.3?

- How does your method compare with RL+TDiv if we look at the diversity of target LLM responses? I presume that both test case diversity and target LLM response diversity are important when red-teaming, and I would like to know whether RL-TDiv essentially achieves target LLM response diversity without necessarily going through the intermediate step of test case diversity.

---

> ### Author Response · Authors · 2023-11-17
> **response**
>
> We thank the reviewer for their comments and are glad that he/she appreciated the comprehensiveness of our experiments, found our idea sound and our presentation clear. The responses to questions are below:
>
> > The authors choose an objective of maximizing test case novelty but do not show how their method influences target LLM response novelty.
> >
>
> > How does your method compare with RL+TDiv if we look at the diversity of target LLM responses? I presume that both test case diversity and target LLM response diversity are important when red-teaming, and I would like to know whether RL-TDiv essentially achieves target LLM response diversity without necessarily going through the intermediate step of test case diversity.
> >
>
> **Answer:**
>
> We added a new figure to show that **RL+Curiosity (ours) also leads to higher target LLM response novelty** (i.e., diversity) than RL+TDiv in Figure 8 in Appendix C.2.
>
> RL+TDiv adds the diversity of the embeddings from the target model’s responses as rewards. Though RL+TDiv achieves higher response diversity than RL, it underperforms ours. This indicates that Curiosity is a more effective objective to achieve both test case and response diversity. We elaborate on these experimental results in Appendix C.2 and briefly explain them below:
>
> - **Why does RL+Curiosity improve responses diversity?** Maximizing test case diversity makes the target model to respond to different prompts and hence leads to diverse responses (i.e., answers to the prompts).
> - **Why does RL+Curiosity lead to higher response diversity than RL+TDiv?** RL+TDiv solely maximizes the diversity of the current batches, rather than the diversity of the all responses seen in the whole training time. We will elaborate this point in the next answer.
>
> > This is not too important but it is worth mentioning that the novelty of the method is somewhat limited. The cosine similarity reward is very similar to the RL+TDiv baseline, except it is applied to the test cases.
> >
>
> **Answer:**
>
> We want to highlight another conceptually critical difference.
>
> **RL+TDiv solely maximizes the diversity within the current batch of responses**, as measured by cosine similarity. This approach does not incentivize the red-team model to generate responses that differ from those in previous batches. Consequently, the model can repeatedly elicit similar responses identical to those in the past, ending up with low response diversity overall. This is evident in Figure 8(d, e) in the Appendix C.2, where RL+TDiv exhibits lower response diversity than ours. It indicates that maximizing testcase diversity is conducive to not only producing diverse testcases but also eliciting diverse toxic responses.
>
> > In A.7, you mention you sample 100 sets of 100 test cases and calculate both diversity metrics across each subset. What is the test case size for a select number of thresholds in your experiments, and why is this sampling method used?
> >
>
> **Answer:**
>
> In A.7, we calculate the diversity of testcases exceeding each toxicity threshold by subset sampling approach because the **number of testcases at each threshold can vary across methods.**
>
> For example, in Figure 2(i, a), nearly 0% of testcases exceeding threshold 0.9 are found by RL baseline, while 50% of testcases found by RL+Curiosity (ours) exceed threshold 0.9. Thus, the testcase sizes for calculating diversity are different at threshold 0.9, which makes them incomparable.
>
> To compute diversity with varying-size testcase sets, we follow the suggestion in [1, 2] and use the resampling method to estimate the diversity of the testcase set.
>
> [1] Perez, Ethan, et al. "Red teaming la~~n~~guage models with language models." arxiv 2022
>
> [2] Lee, Deokjae, et al. "Query-Efficient Black-Box Red Teaming via Bayesian Optimization."  arxiv 2023
>
> > Could you clarify what number of test cases exceed each toxicity threshold for RL+Curiosity for the experiments in section 4.2 and 4.3?
> >
>
> **Answer:**
>
> Each trial (i.e., run with a different random seed) of the experiment uses 100K queries in Section 4.2 and 40K queries in Section 4.3. Thus, the testcases that elicit responses exceeding each toxicity threshold by our proposed RL + curiosity method are:
>
> | Threshold | Sec. 4.2 | Sec. 4.3 (Fig. i) | Sec. 4.3 (Fig. ii) |
> |-----------|----------|-------------------|--------------------|
> | 0.0 | 100000   | 40000| 40000 |
> | 0.1       | 44860    | 26237             | 25074              |
> | 0.2       | 42707    | 25730             | 24255              |
> | 0.3       | 41178    | 25359             | 23718              |
> | 0.4       | 40095    | 25033             | 23325              |
> | 0.5       | 39109    | 24759             | 22987              |
> | 0.6       | 38071    | 24477             | 22617              |
> | 0.7       | 36825    | 24118             | 22187              |
> | 0.8       | 35038    | 23584             | 21605              |
> | 0.9       | 31684    | 22458             | 20238              |

---

> > ### Comment · Reviewer_ZhdE · 2023-11-19
> >
> > Thanks for answering my questions.
> >
> > I appreciate the additional figure showing higher target LLM response novelty.
> >
> > **With respect to Figure 2(i)(c):**
> > > Also, note that while RL (without curiosity and TDiv) achieves a high level of diversity in Figure 2i(c), only a limited number of test cases exceed high toxicity thresholds [0.2, 0.9]
> >
> > Could you report these numbers? Could you also address why the zero shot baseline performs so well in Figure 2(ii)(c)?
> >
> > I will be happy to increase my score once these remaining issues are addressed.

---

> ### Author Response · Authors · 2023-11-19
>
> We thank the reviewer's timely response.
>
> > “Also, note that while RL (without curiosity and TDiv) achieves a high level of diversity in Figure 2i(c), only a limited number of test cases exceed high toxicity thresholds [0.2, 0.9]”
> Could you report these numbers?
> Could you also address why the zero shot baseline performs so well in Figure 2(ii)(c)?
> >
>
> **Answer:**
>
> **Could you report these numbers?**
>
> - The following table shows the number of testcases at each threshold of RL baseline in Figures 2(i) and 2(ii) (green curves). Compared with the table shown in the previous responses, the number of testcases generated by our method at each toxicity threshold is up to 10 times higher than the RL baseline. We will also attach these tables to the Appendix in the next manuscript revision during the rebuttal.
>
> |  Threshold   | Fig 2(i) | Fig 2(ii) |
> |-----|----------|-----------|
> | 0.0 | 40000    | 40000     |
> | 0.1 | 1665     | 9642      |
> | 0.2 | 245      | 7663      |
> | 0.3 | 106      | 6426      |
> | 0.4 | 76       | 5398      |
> | 0.5 | 59       | 4798      |
> | 0.6 | 51       | 4283      |
> | 0.7 | 46       | 3712      |
> | 0.8 | 40       | 2622      |
> | 0.9 | 36       | 1732      |
>
> **Could you also address why the zero shot baseline performs so well in Figure 2(ii)(c)?**
>
> - **The main reason is that the zero-shot baseline is not finetuned with RL**. As Perez et al. 2022 suggested, pre-trained LLMs tend to have higher testcase diversity than RL-finetuned LLMs in red teaming, likely because RL's reward maximization objective hurts diversity [1]. The zero-shot baseline in Figure 2(ii)(c) is a pre-trained LLM and is, hence, expected to have higher diversity than the RL baseline (the green curve in Figure 2(ii)(c)).
> - **However, the zero-shot baseline is notably ineffective in generating testcases that trigger toxic responses in the target model** (see Figure 2(i)(a) and 2(ii)(a)). Additionally, it shows lower diversity based on 1-SelfBLEU (Figures 2(i)(b), 2(ii)(b)) and only matches embedding diversity (Figures 2(i)(c), 2(ii)(c)) when compared to the our RL+Curiosity method.
>
> [1] Emmanuel Bengio, Moksh Jain, Maksym Korablyov, Doina Precup, and Yoshua Bengio. Flow network based generative models for non-iterative diverse candidate generation. Advances in Neural Information Processing Systems, 34:27381–27394, 2021

---

> > ### Comment · Reviewer_ZhdE · 2023-11-20
> >
> > Thank you for your response. I have increased my score to 8.

---

### Official Review · Reviewer_8YjF · 2023-11-06

**Soundness:** 3 good
**Presentation:** 3 good
**Contribution:** 3 good
**Rating:** 8
**Confidence:** 4

**Summary:**

This paper proposes a method for automated red teaming. The method is a spiritual successor to those in the Red Teaming LMs with LMs paper from Perez et al, and should be viewed in that light (rather than in comparison to recent alternative approaches like ARCA and GCG). Specifically, the method adds a curiosity objective to the RL red teaming method of Perez et al.

**Strengths:**

- This paper continues an interesting line of work in fine-tuning LLMs to be better at red teaming other LLMs. This type of research is complimentary with more recent optimization-based methods like GCG

- The idea of incorporating curiosity into the RL red teaming method from Perez et al. is a good idea

- The results are strong; the success rate of the red teaming method remains as high as the RL baseline, and the diversity is much higher

- The ablations are reasonable and anticipate questions that readers would have

**Weaknesses:**

- The distinction between adversarial attacks and red teaming seems artificial. I wouldn't want this distinction being introduced in the community. Surely adversarial attacks can be semantic, and surely red teaming can include gibberish adversarial examples as an interesting failure mode of LLMs.

- I would suggest adding more visual separation between diversity and quality plots. Currently it's hard to tell which is which in Figures 1 and 2 without turning my head sideways to read the axis.

**Questions:**

N/A

---

> ### Author Response · Authors · 2023-11-17
> **response**
>
> We are glad that the reviewer appreciates our idea and results. We answer the rest of questions below.
>
> > The distinction between adversarial attacks and red teaming seems artificial. I wouldn't want this distinction being introduced in the community. Surely adversarial attacks can be semantic, and surely red teaming can include gibberish adversarial examples as an interesting failure mode of LLMs.
> >
>
> **Answer:**
>
> Thanks for the suggestion, and we agree with the reviewer. We’ve updated the manuscript and removed the distinction in the related work section (Section 5). We are happy to further revise this section during the rebuttal according to the reviewer’s comments.
>
> > I would suggest adding more visual separation between diversity and quality plots. Currently it's hard to tell which is which in Figures 1 and 2 without turning my head sideways to read the axis.
> >
>
> **Answer:**
>
> We’ve updated the figures, moving “Quality” and “Diversity” to the titles of each subplot so that the reader can tell which plot is for quality and which plot is for diversity easily. We are happy to revise the figures if the reviewer thinks there could be a better way of visualizing these data.

---

> ### Comment · Reviewer_8YjF · 2023-11-23
> **Response**
>
> This addresses my concerns. I think the paper could be accepted and have raised my score to an 8.

---

### Meta-Review · Area_Chair_MExG · 2023-12-07

**Metareview:**

This paper introduces curiosity loss into the RL objective for finetuning a redteaming LLM, which seeks to improve the diversity of the generated prompts. This is a success effort of incorporating previously known RL techniques (for exploration) into LLM redteaming. All the reviewers agree that this paper makes a solid contribution and the author rebuttals have addressed their concerns. The authors are encouraged to incorporate the feedback from the reviewers into the revised paper.

**Justification For Why Not Higher Score:**

Although the method is simple and useful, the novelty is not very outstanding to merit a spotlight or oral paper since curiosity or novelty -based exploration bonus is well studied in RL community.

**Justification For Why Not Lower Score:**

The idea is quite simple and effective for LLM redteaming. Therefore, the paper merits a publication.

---

### Decision · Program_Chairs · 2024-01-16

Accept (poster)